# Contributions to human breast milk microbiome and enteromammary transfer of *Bifidobacterium breve*

Kattayoun Kordy[1,2☯], Thaidra Gaufin[3‡], Martin Mwangi[3☯], Fan Li[1,2,3☯], Chiara Cerini[1‡], David J. Lee[1☯], Helty Adisetiyo[1‡], Cora Woodward[3‡], Pia S. Pannaraj[1,2‡], Nicole H. Tobin[3☯], Grace M. Aldrovandi[3☯]*

**1** Department of Pediatrics, Children's Hospital of Los Angeles, Los Angeles, CA, United States of America,
**2** Department of Pediatrics, University of Southern California, Los Angeles, CA, United States of America,
**3** Department of Pediatrics, University of California Los Angeles, Los Angeles, CA, United States of America

☯ These authors contributed equally to this work.
‡ These authors also contributed equally to this work.
* GAldrovandi@mednet.ucla.edu

**Data Availability Statement:** Accession Numbers: Sequencing data are available from the NCBI Short Read Archive (SRA) under submission SUB4724831 and BioProject PRJNA295847.

## Abstract

Increasing evidence supports the importance of the breast milk microbiome in seeding the infant gut. However, the origin of bacteria in milk and the process of milk microbe-mediated seeding of infant intestine need further elucidation. Presumed sources of bacteria in milk include locations of mother-infant and mother-environment interactions. We investigate the role of mother-infant interaction on breast milk microbes. Shotgun metagenomics and 16S rRNA gene sequencing identified milk microbes of mother-infant pairs in breastfed infants and in infants that have never latched. Although breast milk has low overall biomass, milk microbes play an important role in seeding the infant gut. Breast milk bacteria were largely comprised of *Staphylococcus*, *Streptococcus*, *Acinetobacter*, and *Enterobacter* primarily derived from maternal areolar skin and infant oral sites in breastfeeding pairs. This suggests that the process of breastfeeding is a potentially important mechanism for propagation of breast milk microbes through retrograde flux via infant oral and areolar skin contact. In one infant delivered via Caesarian section, a distinct strain of *Bifidobacteria breve* was identified in maternal rectum, breast milk and the infant's stool potentially suggesting direct transmission. This may support the existence of microbial translocation of this anaerobic bacteria via the enteromammary pathway in humans, where maternal bacteria translocate across the maternal gut and are transferred to the mammary glands. Modulating sources of human milk microbiome seeding potentially imply opportunities to ultimately influence the development of the infant microbiome and health.

## Introduction

The complex interplay between the microbiome, maternal immune constituents and infant gut colonization is of great importance to the development of the human microbiome, however, the sources of microbes in human milk still require further elucidation. Both culture and non-culture methods have identified aerobic and anaerobic bacterial species in milk, including

**Funding:** This study was supported in part by the NIH K12 Child Health Research Career Development Award (HD052954) to KK. The funder had no role in study design, data collection and analysis, decision to publish, or preparation of the manuscript.

**Competing interests:** I have read the journal's policy and the authors of this manuscript have the following competing interests: Dr. Kordy performed this work while at CHLA and is currently affiliated with Novartis. We confirm that this does not alter our adherence to PLOS ONE policies on sharing data and materials.

strict anaerobes typically compartmentalized in the gut [1–6]. Precolostrum, prior to labor, contains bacterial species similar to milk after labor[6–8]. The same microbes have been found in both milk and feces of mother-infant pairs[5]. Human milk plays an important role in establishing the infant gut microbiome, serving as a source of lactic acid-producing bacteria and human milk oligosaccharides for the infant gut[9, 10]. Similar to murine models[11], human milk and its microbes facilitate differentiation of the neonatal intestinal epithelium, development of the gut associated lymphoid tissue and maturation of the neonatal immune system [12].

Proposed sources for the bacteria in human milk include skin and areolar bacteria, the environment, and infant's oral microbiota through retrograde flow that occurs during nursing[6, 8, 13]. Alterations in the bacterial composition of human milk have been associated with maternal BMI, weight gain, hormones, lactation stage, gestational age, and mode of delivery [13, 14]. Although controversial, the presence of an enteromammary pathway, whereby bacteria, assisted by dendritic cells, translocate across the maternal intestinal mucosa and are delivered to the lactating mammary gland, has been proposed as one source of the bacteria including anaerobes in pre-colostrum and milk[8, 15]. There is some supportive evidence that maternal ingestion of probiotics increases breast milk levels of these microbes[16–18]. If this pathway proves to exists in humans, this suggests that modulation of maternal gut flora may directly impact infant health [15]. While murine [11, 19] and bovine studies [20, 21] suggest that bacteria enter milk from an enteromammary pathway, this is challenging to prove in humans and has been the subject of debate.

Disentangling the contributions of potential sources of bacteria in breast milk is difficult. We sought to assess breastfeeding and potential retrograde flow of bacteria from the infant's oral cavity by performing 16S rRNA gene sequencing on samples from two groups of mother-infant pairs, one in which infants latched onto their mother's breast and a second group of infants that never latched. Furthermore, we investigate the potential role of an enteromammary pathway to the human milk microbiome by performing shotgun metagenomic sequencing in an infant born via Caesarian section. We found that the process of breastfeeding is a potentially important mechanism for propagation of breast milk microbes through retrograde flux via infant oral and areolar skin contact. Our data also implicates a connection between *Bifidobacteria breve* in maternal gut and breast milk suggesting that intestinally-derived bacteria may translocate to the mammary gland and colonize the infant intestine.

## Materials and methods

A subset of mother-infant pairs were selected from a larger cohort who delivered in Los Angeles, California from 2010 to 2014. The Institutional Review Board of Children's Hospital of Los Angeles approved the study and written consent was obtained. Fifteen of the mother-infant pairs latched for breastfeeding and 5 infants who had never latched were selected for comparison. Samples collected included expressed milk, maternal areolar skin swabs, and infant stool samples as previously described [22]. Swab samples were also obtained from the mother's oral mucosa, vagina, and rectum and the infant's buccal mucosa. To capture what the baby was actually exposed to, the study coordinator, wearing standard laboratory gloves, collected one swab (Copan, Murrieta, California, USA) from each maternal areola after the mother performed her typical cleaning, but before the baby latched. After collection, samples were transported on ice, and then were either placed in Stool DNA Stabilizer buffer (Stratec, Berlin, Germany) or frozen 'neat' within 4 hours of collection and stored at -80˚C.

DNA extraction and purification was performed on frozen human milk samples, areolar skin samples, stool samples, and swabs obtained from the oral mucosa, vagina, and rectum as

previously described[22]. Quantitative PCR (qPCR) was used to determine the copies of 16S and GAPDH genes per ng of total DNA extracted from each human milk sample. 16S targeting primers 515F (`GTG YCA GCM GCC GCG GTA A`) and 806R (`GGA CTA CNV GGG TWT CTA AT`) were designed based on Caporaso et al[23] and acquired from Eurofins Genomics (Louisville, KY). GAPDH primers GAPDH-for (`ACC ACA GTC CAT GCC ATC AC`) and GAPDH-rev (`TCC ACC ACC CTG TTG CTG TA`) were acquired from IDT (Skokie, Illinois) as ready-made primers. Quantitation for 16S and GAPDH targets were performed separately in qPCR reactions containing 1x SSO Advanced Universal SYBR Green Supermix (Bio-Rad, Hercules, CA), and 0.5 uM of each paired primer and approximately 1 ng of template DNA. qPCR thermocycling was carried out using a Bio-Rad CFX96 instrument with the following conditions: GAPDH, 98C hold for 2 min followed by 40 cycles of 98C for 20 sec and 60.5C for 40 sec; 16S, 98C hold for 2 min followed by 40 cycles of 98C for 20 sec and 61.C for 40 sec. Standards for GAPDH were obtained by 10-fold serial dilutions of DNA extracted from human T cells and standards for 16S DNA were prepared as described previously[22]. The samples and standards were analyzed in triplicate using the CFx Maestro program (Bio-Rad) and results are reported as the mean log copies/ng of total DNA.

For all 20 subjects, the V4 region of the 16S rRNA gene was amplified and sequenced as previously described[22, 24, 25]. DNA amplicon concentrations were then quantified on a 2100 Bioanalyzer (Agilent Technologies, Santa Clara, California, USA). Pooled libraries were sequenced on an Illumina MiSeq instrument using 2x150bp v2 chemistry [25]. DADA2 version 1.4 was used for error correction, sequence inference, and chimera filtering with default settings. Taxonomic classification was performed using the RDP naïve Bayesian classifier. Contaminant sequence variants were identified as those with at least 10% of their abundance derived from negative control samples and were excluded from all subsequent analyses as previously described[26]. Diversity, ordination, and permutational multivariate analysis of variance (PERMANOVA) analyses were performed using the 'phyloseq' (version 1.22.3) and vegan (version 2.5–2) R packages. PERMANOVA assesses overall microbial variation by measuring the fraction of variance that can be explained by each covariate. Zero-inflated negative binomial (ZINB) regression models were used to test for differential abundance of specific bacterial taxa using rarefied sequence counts as the outcome and clinical covariates as the independent variable. Infant age in days was included as a covariate in all models to account for differences in microbial composition by age. The Benjamini-Hochberg FDR method was used to control for multiple hypotheses and results with an adjusted p-value less than 0.05 were accepted as significant. Source tracking analysis to help determine site contribution to breast milk and infant stool was performed using SourceTracker version 1.0.0 with default parameters and the amplicon sequence variant (ASV) table as input.

Shotgun metagenomic sequencing was performed as previously described[2] on 6 subjects in the latched cohort. Metagenomic libraries were constructed from the previously extracted DNA using the Illumina Nextera XT DNA library preparation kit following manufacturer's instructions. Sequencing was performed on a NextSeq500 platform to a target depth of 5 million reads per sample. Adapter trimming and quality filtering were performed using trim galore, host sequences were removed using kneadData, and taxonomic classification was performed with Kraken (v0.15-beta). ConStrains was used to perform strain-level analysis with parameters 'min-coverage 5'.

## Results

Fifteen mother-infant pairs where the infant latched during breastfeeding and 5 mother-infant pairs whose mothers expressed breast milk but the infants did not latch for medical reasons

were included (Table 1). Maternal age and length of pregnancy were similar between the two groups. However, more mother-infant pairs in the latched group were delivered vaginally (53%) whereas the majority (80%) of the non-latched group underwent a non-elective Cesarean section. More of these never-latched infants (40%) and mothers (80%) received antibiotics than their latched counterparts. In the never latched cohort, 2 maternal-infant pairs received both maternal intrapartum and infant postpartum antibiotics. Their samples were collected during the first 3 weeks of life. The other 2 never latched maternal-infant pairs had maternal intrapartum antibiotics only. They were sampled in the first 3 days of life. In the latched cohort, one mother-infant pair received both maternal intrapartum and infant postpartum antibiotics; the infant's sample was collected at DOL 19. Six additional mothers in the latched group received antibiotics during delivery only and were sampled from DOL 3 to 55.

Of the 20 mother-infant pairs, 15 pairs (13 latched and 2 never-latched) were included in the final analysis. Five pairs were eliminated due to insufficient quantity of qPCR-recovered milk bacteria or DNA and were not true positives by qPCR (1.69–5.22 log 16S V4 copies/ng DNA). Of the included never-latched pairs, 1 subject had 4 different milk samples longitudinally collected within the first two weeks of life which were analyzed individually. In the breastfed group, breast milk bacteria were largely comprised of *Staphylococcus*, *Streptococcus*, *Acinetobacter*, and *Enterobacter* which were primarily derived from areolar skin and infant

**Table 1.  Clinical characteristics of mothers and their infants (n = 20 mother-infant pairs).**

| Demographics of mother-infant pairs | | |
|---|---|---|
| | *Latched (n = 15)* | *Never-latched (n = 5)* |
| Maternal age (years) | 31 (17–38) | 25 (23–46) |
| Length of pregnancy (weeks) | 39 (33–41) | 38 (34–41) |
| Mode of delivery | | |
| Vaginal (%) | 8 (53.3%) | 1 (20%) |
| Elective Cesarean (%) | 5 (33.3%) | 0 (0%) |
| Non-elective Cesarean (%) | 2 (13.3%) | 4 (80%) |
| Maternal antibiotic treatment | | |
| Before delivery[a] | 0 (0%) | 0 (0%) |
| During delivery[b] | 7 (46.7%) | 4 (80%) |
| After delivery | 0 (0%) | 0 (0%) |
| No antibiotic treatment | 8 (53.3%) | 1 (20%) |
| Infant gender (male:female) | 7:8 | 4:1 |
| Infant age (days) | 22 (3–111) | 5 (1–20) |
| Ethnicity | | |
| Hispanic | 10 (66.7%) | 3 (60%) |
| Caucasian | 2 (13.3%) | 1 (20%) |
| Asian | 3 (20%) | 0 (0%) |
| African American | 0 (0%) | 1 (20%) |
| Feeding | | |
| Exclusive breast milk | 6 (40%) | 0 (0%) |
| Mixed (formula + breast milk) | 9 (60%) | 3 (60%) |
| Nothing by mouth | 0 (0%) | 2 (20%) |
| Infant antibiotic treatment | 1 (6.7%) | 2 (40%) |

Data are shown as median, range, or percentage.

[a]During pregnancy until 48 hours before delivery.

[b]During the 48 hours before delivery and in labor.

**Table 2. SourceTracker analysis of the potential sources of the breast milk microbiome.**

| Source[a] | Mean[b] | Median[b] | Standard Deviation[b] | Minimum[b] | Maximum[b] |
|---|---|---|---|---|---|
| Maternal areolar skin | 0.45949 | 0.28783 | 0.39404 | 0.00397 | 0.99885 |
| Infant oral | 0.25818 | 0.08122 | 0.34048 | 0 | 0.93847 |
| Maternal oral | 0.01410 | 0 | 0.03555 | 0 | 0.15192 |
| Maternal rectum | 0.00002 | 0 | 0.00008 | 0 | 0.00037 |
| Maternal vagina | 0.00162 | 0 | 0.00691 | 0 | 0.03242 |
| Unknown | 0.26659 | 0.02206 | 0.37471 | 0 | 0.92936 |

[a]Percentages of the breast milk microbiome is inferred to come from each of these sources: maternal areolar skin, oral, rectum, vagina and infant oral cavity.

[b]Mean, median, standard deviation, minimum, and maximum refer to the distribution of these percentages as it is calculated independently for each mother-infant pair.

oral sites according to SourceTracker analysis (Table 2). Notably, the two mothers with never-latched infants showed different compositions with pure *Staphylococcus* in one and *Staphylococcus*, *Finegoldia* and *Corynebacterium* in the other (Fig 1A).

In a sub-analysis of the latched samples, PERMANOVA identified exclusive breastfeeding as a significant driver of overall microbial variation ($R^2$ = 0.028, p<0.001). No significant differences in diversity or relative abundance of specific bacterial taxa were noted by exclusive breastfeeding, delivery, or sex. Intriguingly, the genus *Bifidobacterium* on 16S rRNA sequencing was found in the breast milk, infant stool, and maternal rectal samples from a single mother-infant pair with Caesarian delivery. We utilized shotgun metagenomics to further resolve the strain identity of this shared bifidobacteria. Species-level analysis showed *Bifidobacterium breve* to be only a minor component of the maternal gut community (0.07% relative abundance) but a significantly larger portion of the breast milk and infant gut microbiomes (28.44% and 67.7% relative abundance, respectively) (Fig 1B). Strain-level mutational profiles also revealed a distinct strain of *Bifidobacterium breve* to be common across these three samples from the same mother-infant pair.

## Discussion

The process of breast feeding plays a critical role in development of the infant gut microbiome. The initial seeding of the infant gut in the first few months of life is necessary for infant immune development and overall health[27–31] with breastfeeding exclusivity and percentage critically influencing the infant gut microbiome[22, 31]. In our analysis, the breast milk and infant microbiomes are seeded through multiple pathways, though primarily from areolar skin and infant oral sites. Additionally, a single mother-infant breastfeeding pair provide intriguing evidence for an enteromammary pathway contributing the same strain of *Bifidobacterium breve* found in maternal intestine and human milk as well as her infant's gut. This infant was delivered via Caesarian section limiting the possibility of infant colonization during delivery. Furthermore, even though *Bifidobacterium breve* constituted less than 1% of the maternal rectal sample, it made up 28% of the maternal milk sample. This single species of bacteria then composed 68% of the infant's gut microbiome.

There is increasing evidence of transfer of anaerobic bifidobacteria from maternal intestine to breast milk then colonizing and expanding in infant gut[5]. Bifidobacteria are amongst the first bacteria to colonize the infant intestine and are associated with decreases in the risk of obesity, asthma, atopy, and all-cause mortality from necrotizing enterocolitis in pre-term infants[27, 32, 33]. Given the importance of bifidobacteria in infant health, it is logical for mothers to selectively enrich and support colonization by this bacterial population.

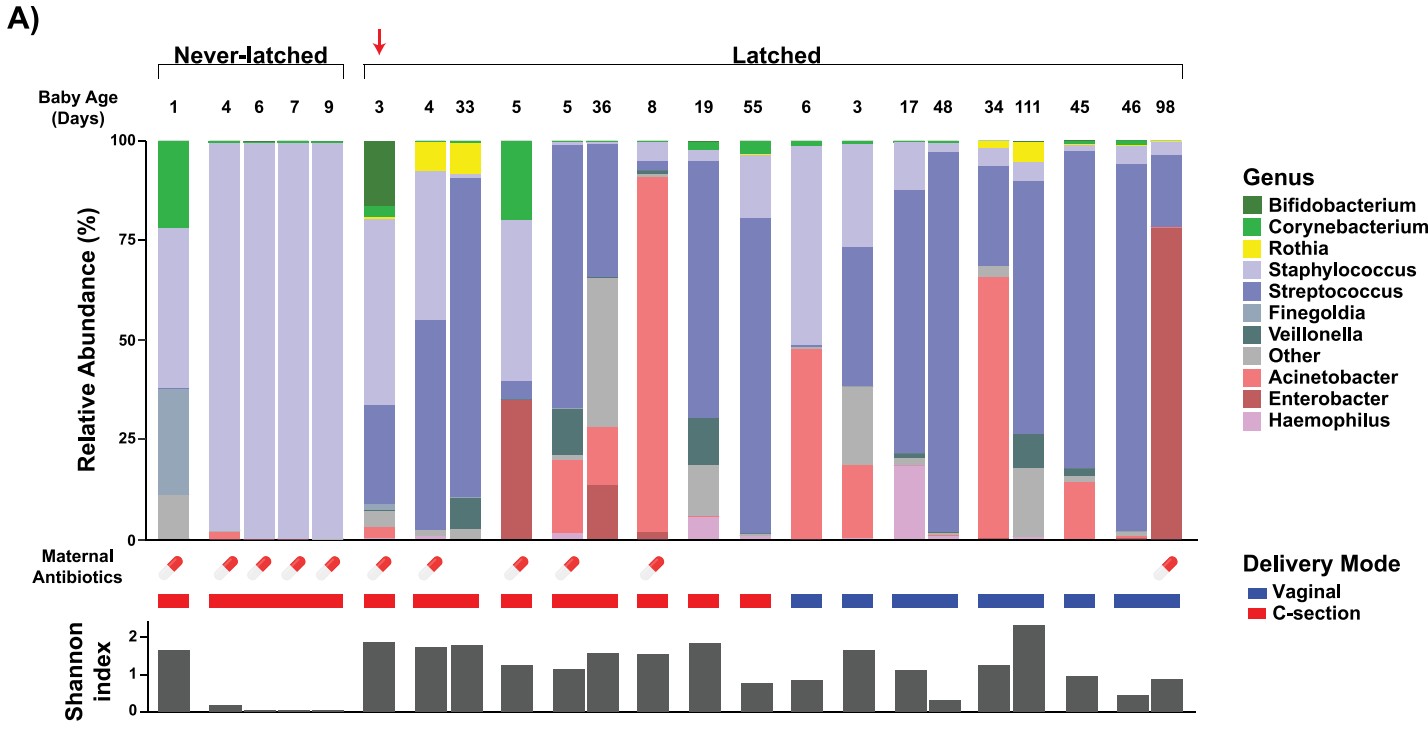

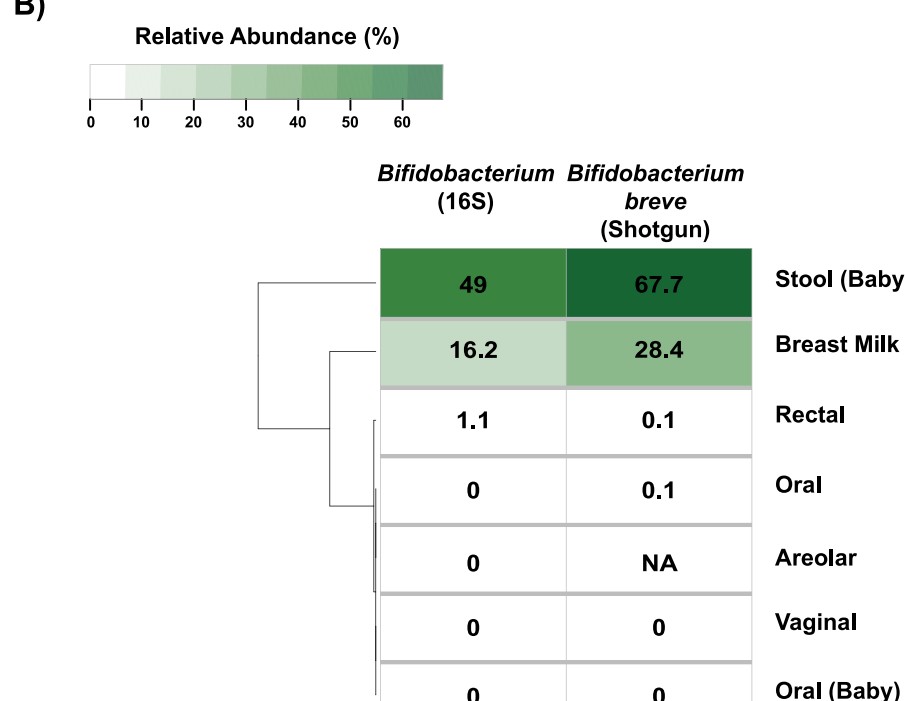

**Fig 1. Microbiome composition of human milk samples.** (A) Infant age (days) at time of sampling, relative abundance, maternal antibiotics in the 14 days prior to sampling, mode of delivery, and Shannon diversity of human milk samples from mothers with infants who either have latched or never latched. Samples from the same mother collected on different days are grouped. Milk from mothers who never had their infants latched were dominated by *Staphylococcus* in one and *Staphylococcus*, *Finegoldia and Corynebacterium* in the other. Note the absence of *Streptococcus* and lower overall diversity of never-latched samples. In contrast, samples from mothers with latched infants, also born via Caesarian section in the first 10 days of life (n = 5), contained *Streptococcus*, *Acinetobacter*, and *Enterobacter* in addition to

*Staphylococcus*. (B) Relative abundance of genus *Bifidobacterium* by targeted 16S rRNA gene sequencing (left) and shotgun metagenomics (right) in a single milk sample (arrow) shown in Panel A. *Bifidobacterium breve* appears to be selectively cultivated in the mother's milk and then makes up the majority of her infant's early gut microbiome.

Milk ducts are bidirectional channels [34] so it is likely that bacteria from skin and the infant oral cavity populate human milk. Furthermore, there is recent support that strains of bacteria found in precolostrum may have a significant impact in the initial establishment of the infant oral microbiota [6]. Our analysis is also suggestive of the role of retrograde oral seeding of bacteria into maternal milk from the act of infant suckling. Both mother-infant pairs with sufficient data where the infant never latched had a predominance of skin flora consisting mostly of *Staphylococc*us and some *Corynebacterium* with a notable absence of *Streptococcus*. In contrast, most of the milk samples from latching pairs had at least some and often a majority of *Streptococcus* present in their milk samples. Latched samples also had a greater overall diversity including *Acinetobacter*, *Enterobacter*, *Veillonella*, and *Haemophilus* in addition to the *Staphylococcus*, *Streptococcus* and *Corynebacterium*, consistent with previous studies[13]. However, with only 2 of the never-latched mothers having sufficient bacteria present in their milk for analysis, our data are insufficient to draw any definitive conclusions about the role of latching on milk microbe composition.

Our study is limited by the small sample size, the range of gestational ages of the infants at birth, the range of ages at sample collection, and by the inability to show directionality of bacterial transfer. Some microbes found on areolar skin are also present on the mucosal surfaces of the gastrointestinal tract[8] and our methods do not determine the source of these microbes in the breast milk. Prior investigations have demonstrated an enteromammary pathway in animals [8, 15] and a viable strain of *Bifidobacterium breve* in maternal faeces, breast milk and neonatal faeces in vaginally delivered infants [5]. Although our report suggest evidence for an enteromammary pathway by finding a single strain of *Bifidobacterium breve* in maternal rectum, breastmilk and infant stool, there is a possibility that maternal fecal microbes can be spread by the mother herself to the skin and breast although this is less likely in an infant delivered via Caesarian section. More definitive evidence is required to support the role of an enteromammary pathway in humans for translocating critical microbial communities to the breast milk compartment and eventually seeding the infant gut via breastfeeding. Our findings need to be investigated with larger cohorts and molecular-based surveys or culture-based analyses to validate the shotgun metagenomics data.

In conclusion, our data suggests that the process of breastfeeding and interaction between areolar skin and infant oral cavity are potentially critical for seeding the milk microbiome. Furthermore, our report provides intriguing evidence suggestive of an enteromammary pathway in humans with transfer of a single strain of *Bifidobacterium breve* in maternal intestine, breastmilk and infant stool in an infant delivered via Caesarian section. These sources of milk microbiome seeding, if verified in larger studies, may support opportunities to modulate bacteria found in human breast milk and ultimately development of the infant microbiome.

## Acknowledgments

The authors are grateful to the volunteers, their families, and their medical providers for participating in this study. Special thanks to Amit Oberai for his assistance with the initial analysis of this data.

## Author Contributions

**Conceptualization:** Kattayoun Kordy, Grace M. Aldrovandi.

**Data curation:** Martin Mwangi, Fan Li, Chiara Cerini, Helty Adisetiyo.

**Formal analysis:** Kattayoun Kordy, Thaidra Gaufin, Martin Mwangi, Fan Li, David J. Lee, Cora Woodward, Nicole H. Tobin, Grace M. Aldrovandi.

**Funding acquisition:** Kattayoun Kordy.

**Investigation:** Kattayoun Kordy, Martin Mwangi, Pia S. Pannaraj, Grace M. Aldrovandi.

**Methodology:** Kattayoun Kordy, Fan Li, Chiara Cerini, David J. Lee, Helty Adisetiyo, Cora Woodward, Grace M. Aldrovandi.

**Project administration:** Kattayoun Kordy, Thaidra Gaufin, Chiara Cerini, Pia S. Pannaraj, Nicole H. Tobin, Grace M. Aldrovandi.

**Resources:** Pia S. Pannaraj.

**Supervision:** Kattayoun Kordy, Nicole H. Tobin, Grace M. Aldrovandi.

**Visualization:** Fan Li, David J. Lee, Nicole H. Tobin.

**Writing – original draft:** Kattayoun Kordy, Martin Mwangi.

**Writing – review & editing:** Kattayoun Kordy, Thaidra Gaufin, Fan Li, Nicole H. Tobin, Grace M. Aldrovandi.

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
