## [Decision Letter · Decision Letter 0]

11 Aug 2019

PONE-D-19-16931

Contributions to human breast milk microbiome and enteromammary transfer of Bifidobacterium breve

PLOS ONE

Dear Dr. Kordy,

Thank you for submitting your manuscript to PLOS ONE. After careful consideration, we feel that it has merit but does not fully meet PLOS ONE’s publication criteria as it currently stands. Therefore, we invite you to submit a revised version of the manuscript that addresses the points raised during the review process.

We would appreciate receiving your revised manuscript by Sep 25 2019 11:59PM. To enhance the reproducibility of your results, we recommend that if applicable you deposit your laboratory protocols in protocols.io, where a protocol can be assigned its own identifier (DOI) such that it can be cited independently in the future. For instructions see: http://journals.plos.org/plosone/s/submission-guidelines#loc-laboratory-protocols

We look forward to receiving your revised manuscript.

Kind regards,

Juan J Loor

Academic Editor

PLOS ONE

Journal Requirements:

1. Our internal editors have looked over your manuscript and determined that it is within the scope of our The Microbiome Across Biological Systems Call for Papers. This collection of papers is headed by a team of Guest Editors for PLOS ONE: Zaid Abdo, Colorado State University, USA; Sanjay Chotrimall, Lee Kong Chian School of Medicine, Nanyang Technological University, Singapore; Noelle Noyes, University of Minnesotta, USA;  Pankaj Trivedi, Colorado State University, USA; and Thomas Dawson, A*STAR, Singapore. The Collection will encompass a diverse range of research articles about microbiomes and human health, the natural and built environment, and new technologies used to study microbiomes. Additional information can be found on our announcement page: https://collections.plos.org/s/microbiome. If you would like your manuscript to be considered for this collection, please let us know in your cover letter and we will ensure that your paper is treated as if you were responding to this call. If you would prefer to remove your manuscript from collection consideration, please specify this in the cover letter.

2. Thank you for including your competing interests statement; "I have read the journal's policy and the authors of this manuscript have the following competing interests: Dr. Kordy performed this work while at CHLA and is currently affiliated with Novartis."

Reviewers' comments:

Reviewer's Responses to Questions

**Comments to the Author**

1. Is the manuscript technically sound, and do the data support the conclusions?

Reviewer #1: No

2. Has the statistical analysis been performed appropriately and rigorously? 

Reviewer #1: Yes

3. Have the authors made all data underlying the findings in their manuscript fully available?

Reviewer #1: Yes

4. Is the manuscript presented in an intelligible fashion and written in standard English?

Reviewer #1: Yes

5. Review Comments to the Author

Reviewer #1: The manuscript describes differences in human milk microbiota composition of mothers whose infants latched and those that had never latched. Data are also presented which suggest vertical transmission of a particular species of Bifidobacterium from the maternal gut to the infant gut by way of human milk. While some of the data are interesting, they seem incomplete. The differences in terms of gestational age and age when samples were collected across infants included in the study are also concerning.

Line 63-64: need citations

Line 63-65: Line is confusing. The existence of an enteromammary pathway is independent of seeding the infant gut. The enteromammary pathway is from the maternal gut to the mammary gland. It sounds as though you mean that vertical transmission is difficult to prove

Line 75: check the formatting of you bacterial names throughout the paper. This should either be bifidobacteria or Bifidobacterium breve.

Lines 85-87: The reference provided doesn’t offer much about information about collection processes, and there isn’t enough provided in the manuscript. Was the breast cleaned in any way before collection? How many swabs were used for each sample, and what type of swabs were used? Did mothers wear sterile gloves? Were the swabs that were not frozen immediately refrigerated?

Line 126: No SourceTracker results are presented in the manuscript. This does not need to be included in the methods if no results are presented. If you do choose to include it, what type of data did you put into the program – an OTU table or rarefied sequences?

Line 144: How close to sample collection did the infants receive antibiotics?

Table 1: Multiple of these infants were pre-term. Human milk and infant microbiota composition are different in mother-infant pairs born pre-term. Milk microbiota has also been shown to differ by lactation stage. Therefore, it doesn’t seem appropriate to compare milk microbiota of mothers who have infants that are 3 and 100 days old. How do you justify the vast differences in both gestational age as well as age in days of infants?

Line 153-154: Did you pool the 4 longitudinal milk samples?

Line 172: What do you mean by overall microbial variation? This contradicts the next sentence, which says that exclusive breastfeeding did not impact diversity or taxa abundance. Furthermore, was there a difference in age of mixed-feeding and exclusive breastfeeding infants? There is a chance that length of breastfeeding, but not proportion of breastfeeding, is influencing composition (i.e. the infant had been suckling for longer).

Line 175-176: Was Bifidobacterium found in these samples from other pairs?

Line 187: The bacteria are not “secreted” into the milk. Please change this wording. Bifidobacterium also likely makes up such a large part of the infant microbiome due to competitive advantage.

Line 196-198: What data are you presenting that support that the milk and infant gut microbiota are seeded through multiple pathways? SourceTracker may have been a good way to demonstrate this but no analysis is included.

Line 200: Please change the words “milk compartment”. Human milk is what was analyzed.

Line 202-204: The difference in abundance of B. breve between the sample types could simply be due to the difference in bacterial load in stool and human milk. Can you elaborate on the statement that B. breve is “selected for” – what exactly do you mean by that?

Line 212: Again, please change wording. Bacteria cannot be secreted.

Figure 1 B: why are other body sites represented in this figure besides maternal and infant stool and breast milk if they are not discussed at all in the results section?

6. PLOS authors have the option to publish the peer review history of their article (what does this mean?). If published, this will include your full peer review and any attached files.

Reviewer #1: No

---

## [Author Response · Author response to Decision Letter 0]

17 Oct 2019

Reviewer #1: The manuscript describes differences in human milk microbiota composition of mothers whose infants latched and those that had never latched. Data are also presented which suggest vertical transmission of a particular species of Bifidobacterium from the maternal gut to the infant gut by way of human milk. While some of the data are interesting, they seem incomplete. The differences in terms of gestational age and age when samples were collected across infants included in the study are also concerning.

Line 63-64: need citations

We have added the following citations:

Oikonomou G, Machado VS, Santisteban C, Schukken YH, Bicalho RC. Microbial diversity of bovine mastitic milk as described by pyrosequencing of metagenomic 16s rDNA. PLoS One. 2012;7(10):e47671.

Contreras GA, Rodriguez JM. Mastitis: comparative etiology and epidemiology. Journal of mammary gland biology and neoplasia. 2011;16(4):339-56.

Gaboriau-Routhiau V, Rakotobe S, Lecuyer E, Mulder I, Lan A, Bridonneau C, et al. The key role of segmented filamentous bacteria in the coordinated maturation of gut helper T cell responses. Immunity. 2009;31(4):677-89.

Ivanov, II, Atarashi K, Manel N, Brodie EL, Shima T, Karaoz U, et al. Induction of intestinal Th17 cells by segmented filamentous bacteria. Cell. 2009;139(3):485-98.

Line 63-65: Line is confusing. The existence of an enteromammary pathway is independent ofseeding the infant gut. The enteromammary pathway is from the maternal gut to the mammary gland. It sounds as though you mean that vertical transmission is difficult to prove

This section has been rewritten for clarity, please see revised draft.

Line 75: check the formatting of you bacterial names throughout the paper. This should either be bifidobacteria or Bifidobacterium breve.

Where appropriate, we specified “genus Bifidobacterium”; species “Bifidobacterium breve”; plural Bifidobacteria. 

Lines 85-87: The reference provided doesn’t offer much about information about collection processes, and there isn’t enough provided in the manuscript. Was the breast cleaned in any way before collection? How many swabs were used for each sample, and what type of swabs were used? Did mothers wear sterile gloves? Were the swabs that were not frozen immediately refrigerated?

Key elements of the collection process as described below have now been summarized in the methods of the manuscript. 

There was no study protocol request for specific cleaning before collection. However, the mother would perform her typical cleaning as if she were to pump for her baby. The rationale behind this was to attempt to capture what the infant was actually exposed to.

One swab per areola for a total of two swabs per patient were collected before the baby latched. The areolar swab kit contained: two (2) snap-tip sterile swabs and two (2) prefilled tubes of transport buffer.

Areolar swab samples were usually taken by the study coordinator who wore standard laboratory gloves, but not surgical sterile gloves. The areolar skin was swabbed in a circular motion starting from the nipple spiraling outwards, avoiding the nipple and the breast skin, for approximately 10 seconds.

Swabs were kept on ice during transport to the laboratory and frozen immediately. 

Line 126: No SourceTracker results are presented in the manuscript. This does not need to be included in the methods if no results are presented. If you do choose to include it, what type of data did you put into the program – an OTU table orrarefied sequences?

In the Results section, we refer to SourceTracker results in the text and have now also included a Table 2 with the SourceTracker findings for additional details. We have updated the methods to clarify that the amplicon sequence variant (ASV) table was used as input to SourceTracker.

Line 144: How close to sample collection did the infants receive antibiotics?

This has been clarified in the re-worked section to include details from the clinical data that was collected. Where applicable, we have included maternal intrapartum antibiotics given during delivery and infant postpartum antibiotics in the latched and never latched cohorts as well as when samples were collected. Unfortunately, the duration of antibiotic treatment given to infants post-delivery was not specified. 

Table 1: Multiple of these infants were pre-term. Human milk and infant microbiota composition are different in mother-infant pairs born pre-term. Milk microbiota has also been shown to differ by lactation stage. Therefore, it doesn’t seem appropriate to compare milk microbiota of mothers who have infants that are 3 and 100 days old. How do you justify the vast differences in both gestational age as well as age in days of infants?

We agree with the comments that there are differences in milk microbiota based on gestational age and age post-gestation. For the purposes of our analyses, we wanted to pursue the specific question of potential sources of milk microbes between latched vs never-latched infants, so we grouped the totality of evidence from each of these cohorts based on their breastfeeding status. Please note in Table 1 that the latched and never-latched infants ranged from minimum 33 and 34 to maximum 41 and 41 weeks with a median of 39 and 38 weeks gestation, respectively. Indeed, with a larger cohort, stratification into the gestational and post-gestational age-specific demographics would be valuable. Unfortunately, with the small sample size, we could not further stratify meaningfully. Additionally, we show the age of the infant in days in Figure 1 so that the data and its limitations are readily apparent to all readers. We have added this concern as a limitation of our study in lines 249-250. 

Line 153-154: Did you pool the 4 longitudinal milk samples?

The four longitudinal milk samples were not pooled. This is clarified in the text: “Of the included never-latched pairs, 1 subject had 4 different milk samples longitudinally collected within the first two weeks of life which were analyzed individually.”

Line 172: What do you mean by overall microbial variation? This contradicts the next sentence, which says that exclusive breastfeeding did not impact diversity or taxa abundance. Furthermore, was there a difference in age of mixed-feeding and exclusive breastfeeding infants? There is a chance that length of breastfeeding, but not proportion ofbreastfeeding, is influencing composition (i.e. the infant had been suckling for longer).

Overall microbial variation was assessed by PERMANOVA, which measures the fraction of variance that can be explained by each covariate. We have clarified this in the methods and results sections. It is possible for a covariate (e.g. exclusive breastfeeding) to explain a significant proportion of variance in microbiome composition but not be associated with significant differences in either diversity or relative abundances.

Duration of breastfeeding may indeed influence the breast milk composition, but we are not able to tease out the role of duration from proportion of breastfeeding given the way the study was designed. Studies that do support breastfeeding exclusivity and percentage influencing the infant gut microbiome (Pannaraj et al, 2017; Ho et al , 2018) have not separated duration from proportion to the best of our knowledge. This would be interesting to investigate in a future, carefully designed study.

Line 175-176: Was Bifidobacterium found in these samples from other pairs?

Only a single mother and her infant had Bifidobacterium (genus)/ Bifidobacterium breve (species) identified in various samples. 

Line 187: The bacteria are not “secreted” into the milk. Please change this wording. Bifidobacterium also likely makes up such a large part of the infant microbiome due to competitive advantage.

We have changed “secreted” to “cultivated” or "enriched". 

Line 196-198: What data are you presenting that support that the milk and infant gut microbiota are seeded through multiple pathways? SourceTracker may have been a good way to demonstrate this but no analysis is included.

We have now included a Table 2 with the SourceTracker results. 

Line 200: Please change the words “milk compartment”. Human milk is what was analyzed.

We have made the suggested change.

Line 202-204: The difference in abundance of B. breve between the sample types could simply be due to the difference in bacterial load in stool and human milk. Can you elaborate on the statement that B. breve is “selected for”– what exactly do you mean by that?

We have simplified the text given that many biological factors could account for the differences and we did not specifically explore the causes for this in our study: “Furthermore, even though Bifidobacterium breve constituted less than 1% of the maternal rectal sample, it made up 28% of the maternal milk sample.”

Line 212: Again, please change wording. Bacteria cannot be secreted.

We have changed “secreted” to “cultivated” or "enriched". 

Figure 1 B: why are other body sites represented in this figure besides maternal and infant stool and breast milk if they are not discussed at all in the results section?

The SourceTracker analysis was performed using data from the sample sites included in this figure. We have clarified this in the results section by adding Table 2 describing the SourceTracker results.

---

## [Decision Letter · Decision Letter 1]

12 Nov 2019

Contributions to human breast milk microbiome and enteromammary transfer of Bifidobacterium breve

PONE-D-19-16931R1

Dear Dr. Kordy,

We are pleased to inform you that your manuscript has been judged scientifically suitable for publication and will be formally accepted for publication once it complies with all outstanding technical requirements.

With kind regards,

Juan J Loor

Academic Editor

PLOS ONE

Additional Editor Comments (optional):

Reviewers' comments:

Reviewer's Responses to Questions

**Comments to the Author**

1. If the authors have adequately addressed your comments raised in a previous round of review and you feel that this manuscript is now acceptable for publication, you may indicate that here to bypass the “Comments to the Author” section, enter your conflict of interest statement in the “Confidential to Editor” section, and submit your "Accept" recommendation.

Reviewer #1: All comments have been addressed

2. Is the manuscript technically sound, and do the data support the conclusions?

Reviewer #1: Yes

3. Has the statistical analysis been performed appropriately and rigorously? 

Reviewer #1: N/A

4. Have the authors made all data underlying the findings in their manuscript fully available?

Reviewer #1: Yes

5. Is the manuscript presented in an intelligible fashion and written in standard English?

Reviewer #1: Yes

6. Review Comments to the Author

Reviewer #1: All comments from the reviewer have been addressed. The only revision would be to have the legend of Figures 1A and 1B together rather than separated by a paragraph.

7. PLOS authors have the option to publish the peer review history of their article (what does this mean?). If published, this will include your full peer review and any attached files.

Reviewer #1: No

---

## [Editor Report · Acceptance letter]

20 Nov 2019

PONE-D-19-16931R1 

Contributions to human breast milk microbiome and enteromammary transfer of *Bifidobacterium breve*

Dear Dr. Kordy:

I am pleased to inform you that your manuscript has been deemed suitable for publication in PLOS ONE. Congratulations! Your manuscript is now with our production department. 

With kind regards,

on behalf of

Dr. Juan J Loor 

Academic Editor

PLOS ONE